Atlantic salmon (Salmo salar) require increased dietary levels of B-vitamins when fed diets with high inclusion of plant based ingredients

Hemre Gro-Ingunn ghe@nifes.no 1
Lock Erik-Jan 1
Olsvik Pål Asgeir 1
Hamre Kristin 1
Espe Marit 1
Torstensen Bente Elisabeth 1
Silva Joana 2
Hansen Ann-Cecilie 1
Waagbø Rune 1
Johansen Johan S. 3
Sanden Monica 1
Sissener Nini H. 1
1 National Institute of Nutrition and Seafood Research (NIFES) , Bergen , Norway
2 Biomar , Trondheim , Norway
3 GIFAS , Inndyr , Norway
Esteban María Ángeles
Electronic publication date: 2016 Sep 29
Publication date: 2016
Volume: 4
Electronic Location ID: e2493
Received 2016 Jun 15; Accepted 2016 Aug 26
Copyright: ©2016 Hemre et al.
Copyright year: 2016
Copyright holder: Hemre et al.
License: This is an open access article distributed under the terms of the Creative Commons Attribution License, which permits unrestricted use, distribution, reproduction and adaptation in any medium and for any purpose provided that it is properly attributed. For attribution, the original author(s), title, publication source (PeerJ) and either DOI or URL of the article must be cited.
License URL: https://creativecommons.org/licenses/by/4.0/

Keywords: Atlantic salmon, B-vitamin recommendations, Plant based feed

Funding: Advanced Research Initiatives for Nutrition and Aquaculture FP7-288925 This study was a part of the ARRAINA; Advanced Research Initiatives for Nutrition and Aquaculture, (7th Framework Programme, FP7-288925, CP-IP Large-scale integrating project). The funders had no role in study design, data collection and analysis, decision to publish, or preparation of the manuscript.

==============================
Aiming to re-evaluate current recommendations for nutrient supplementations when Atlantic salmon are fed diets based on plant ingredients, two regression experiments, with parr and post-smolt, were conducted. A control diet was included to evaluate if ingredients supplied sufficient nutrients without any added nutrient package (NP). The nutrient package consisted of vitamins B, C, E, minerals, cholesterol, methionine, taurine and histidine. This paper focus on B-vitamins. In parr, growth, health and welfare parameters responded on NP additions, but this was not observed in the seawater stage. During three months of feeding, parr tripled their weight. Parr given diets added the NP above NRC (2011) showed improved protein retention, and reduced liver and viscera indices. Post-smolt fed the same diets during five months showed a doubling of weight, but did not respond to the variation in NP to the same extent as parr. Significant regressions were obtained in body compartments for several of the B-vitamins in the premix. Whole body biotin concentration was unaffected by micronutrient premix level, and mRNA expression of the enzymes dependent of biotin showed only weak increases with increased biotin. Muscle thiamine plateaued at a diet level similar to NRC (2011) recommendation in freshwater, and showed stable values independent on premix addition in seawater. The mRNA expression of the enzyme G6PDH (glucose-6-phosphate dehydrogenase) is sensitive to thiamine availability; results did not indicate any need to add thiamine above levels recommended for fish in general. Niacin showed a steady increase in whole body concentrations as feed niacin increased. Muscle riboflavin peaked at a diet level of 12.4 mg kg−1. Sufficient riboflavin is important to avoid e.g., development of cataract. Cataract was not registered to be any problem, neither in fresh- nor in seawater. Cobalamin (B 12) in muscle and liver was saturated at 0.17 mg kg−1 diet. Muscle pyridoxine showed a dose-dependent level in muscle, and peaked around 10 mg kg −1 diet. White muscle ASAT (asparagine amino transferase) activity steadily increased, with indications of stable values when dietary pyridoxine was around 10–16 mg kg −1 diet. Pantothenic acid increased in gill tissue up to a level of 5.5 mg kg −1 soft gill tissue; at a dietary level of 22 mg kg−1. Improved performance, and coverage of metabolic need for niacin was at a dietary level of 66 mg kg −1, riboflavin 10–12 mg kg−1, pyridoxine 10 mg kg−1 and panthotenic acid 22 mg kg−1. Based on these results, recommended B-vitamin supplementation in plant based diets for Atlantic salmon should be adjusted.

Introduction

Atlantic salmon (Salmo salar) diets with a high inclusion of plant ingredients deviate substantially in nutrient composition from diets based on fishmeal. It has been well documented how to balance the amino acid profile and reduce anti-nutritional factors to maintain high growth rates when using high plant protein inclusion diets (Espe et al., 2006; Espe et al., 2007; Krogdahl, Hemre & Mommsen, 2005) and that up to 100% vegetable oil inclusion does not compromise Atlantic salmon growth when oils are blended to give a balanced fatty acid composition (Torstensen et al., 2008). However, minor attention has been given to micronutrient recommendations following the shift towards higher levels of plant ingredients.

Both level and form of B-vitamins and some of the indispensable amino acids, are significantly different in plant ingredients and fishmeal (Hansen et al., 2007; Hansen, Waagbø & Hemre, 2015). B-vitamins are co-factors in the intermediate metabolism of protein, carbohydrates and lipids, and high concentrations of specific B-vitamins are often present in metabolically active tissues (Waagbø, 2010). Previously, when fish meal was the main protein source, B-vitamins were in surplus, thus diminishing the consequences of B-vitamin degradation caused by feed production processes. However, since plant protein ingredients contain lower levels of B-vitamins and/or chemical forms with lower bioavailability, knowledge on recommended B-vitamin levels added to feed prior to processing is needed.

Based on available information (Hertrampf & Piedad-Pascual, 2000; NRC, 2011), the level of riboflavin, vitamin B12 and niacin are low in plant protein materials compared to fishmeal. Corn gluten is low in pantothenic acid and pea meal is low in vitamin B6; all these plant ingredients are readily used in fish diets. In addition, plants are low in methionine and lack taurine. Taurine is not regarded as an indispensable amino acid, but is present in large quantities in marine feed ingredients and considered important in the prevention of fatty liver in animals (Kerai et al., 1998).

For many of the B-vitamins, no recommendations exist for Atlantic salmon. General symptoms for B-vitamin deficiency are reduced growth and appetite (Hansen, Waagbø & Hemre, 2015). Therefore, the reduced growth when salmon were given a high combined plant protein and vegetable oil-based diet (Torstensen et al., 2008), might partly be explained by very low levels of several of the B-vitamins or other components, such as taurine, which are absent in plant ingredients (Espe et al., in press). The accumulation of fat in the liver and around internal organs reported by Torstensen, Espe & Stubhaug (2011) and Espe et al. (2016) when fish meal and fish oil were replaced by high levels of plant ingredients may indicate sub-optimal or deficient levels of micronutrients. This was further supported by Liland et al. (2013a) and Liland et al. (2013b) who found no difference in lipid deposition in Atlantic salmon fed diets high in plant ingredients when extra micronutrients were added to compensate for the low fish meal and fish oil inclusion.

Generally, early stages of development are considered to be more sensitive to nutrient deficiencies than adult stages, mostly due to fish gut immaturity, rapid growth and development and less vitamin storage capacity. It has therefore been hypothesised that parr in freshwater might need higher levels of some of the B-vitamins than post-smolt held in seawater (Waagbø, 2010).

In most of the experiments reported in NRC (2011) fish were fed cold-extruded pellets or semi-purified diets, while the extrusion technique used in modern diets use high temperature and pressure and are harsh towards potentially unstable nutrients. Unfortunately, the majority of vitamin B studies published lack the analytical confirmation of the B-vitamin level in feed and organs (NRC, 2011). This is particularily relevant when feed has been produced by modern extrusion, since water soluble vitamins may have been destroyed resulting in a lower dietary level in the final feed (Barrows et al., 2008). For some of the B-vitamins, up to 40% loss was registered (Barrows et al., 2008). Compensating for the processing loss, and ending with levels close to NRC (2011) recommendations, rainbow trout (Oncorhynchus mykiss) responded with improved metabolism and health, but with no effect on growth (Barrows et al., 2008).

This study is part of the larger EU-funded Arraina project, which has the aim to establish updated micronutrient recommendations for several aquacultured fish species, including Atlantic salmon, when fed diets with high plant ingredient levels. Here, we report the amino acid and B-vitamin recommendations based on two full regression studies: one with Atlantic salmon parr in freshwater (trial 1), and one with Atlantic salmon post-smolt in seawater (trial 2). Both studies were designed in a dose–response manner with the exact same diet-design, using seven diets with graded levels of a nutrient package (NP) added to a basic diet high in plant ingredients. The rationale was to identify if small freshwater parr and larger seawater post-smolt required a similar nutrient package added to their diets. The NP contained essential vitamins, minerals, cholesterol and amino acids (totally 25 nutrients), where the 100% NP was close to NRC (2011) recommendations for salmonids (mainly rainbow trout). The inclusion of the NP ranged from no addition to four fold the 100% addition, and where the 100NP group represented the current knowledge at the time and NRC (2011) for salmonids. This paper presents updated recommendations for the B-vitamins biotin, folate, niacin, pantothenic acid, pyridoxine, riboflavin, thiamine and cobalamin in plant ingredient based diets for Atlantic salmon.

Data from the same dietary experiments on selected minerals and vitamin A, D and K related to bone metabolism and on vitamin C, E and selenium related to redox regulation are given in K Hamre et al. (2016, unpublished data).

Materials and Methods

Experimental diets

The seven experimental diets were produced at Biomar Process Innovation Technical Centre (Brande, Denmark) at appropriate pellet sizes to satisfy gape size of salmon parr (2 mm) and post-smolts (6 mm). Production equipment was modern extrusion, by means of Clextral BC 45, twin screw, 7-sectons, where feed production temperature reached 115°C. All feed analyses were performed on finished extruded feed, so the values given in Tables 1 and 2 are close to similar to the levels given to the fish. All diets contained the same basal mixture of ingredients (Table 1), while a nutrient package (NP) added in graded amounts, replacing some of the field peas. Diets and experimental design were the same in trial 1 with parr in freshwater and trial 2 with post-smolt in seawater. Due to changes in protein and lipid levels between the two size groups; diets in trial 2 held higher lipid and a bit lower protein, these values are given in parenthesis in Table 2. All diets had a low content of marine ingredients, with 8% FM and 2.4% krill meal (of the total recipe), while the remainder of the protein was provided by plant sources. Capelin oil constituted 20% of the added oil, while the remainder was a mix of rapeseed oil, palm kernel oil and linseed oil (55:30:15). Phosphate, lysine, threonine and choline were added to all diets in equal amounts. An antioxidant mixture to protect the feed during production, and yttrium for later digestibility measurements were also added to all diets. Diet acronyms were as follows: 0NP had no addition of the micronutrient premix, then the NP was added in graded amounts to the six diets called 25NP, 50NP, 100NP, 150NP, 200NP and 400NP. The general idea was that the 100NP diets should contain 100% of the assumed requirement (based on available data, primarily for rainbow trout (NRC, 2011) for each nutrient. The 25NP would cover 25% and 400NP would cover 400% of earlier estimated requirement for salmonids. However, the accuracy of this will vary between the different nutrients, due to different amounts already provided by the basic ingredients in the feed formulations, in many cases fulfilling much more that 25% of the assumed requirement. The NP contained vitamin D3, α-tocopherol-acetate, vitamin K3, vitamin A1, ascorbyl monophosphate, vitamin B6, biotin, cobalamin, folate, pantothenic acid, riboflavin, thiamine, niacin, selenium (as inorganic sodium selenite), iodine, copper, cobalt, manganese, zinc, crystalline DL-methionine, and taurine. Crystalline L-histidine and cholesterol were also added in graded amounts. The analyzed composition of the diets is found in Table 2, including proximate composition and micronutrient content.

Table 1 Feed formulation trial 1, and trial 2 (given in brackets when deviating from trial 1).

Nutrient premix (NP), methionine, taurine and cholesterol were added to the diets in graded amounts, and balanced by reducing the content of field peas in the diets; all other ingredients were used in equal amounts in all diets. Data are in g kg−1.

Composition	0NP	25NP	50NP	100NP	150NP	200NP	400NP	
Fish meal SA 68 superprime	80	80	80	80	80	80	80	
Krill meal	24.2	24.2	24.2	24.2	24.2	24.2	24.2	
SPC 60%	180	180	180	180	180	180	180	
Corn gluten 60	40	40	40	40	40	40	40	
Pea protein 75	124 (130)	124 (130)	124 (130)	124 (130)	124 (130)	124 (130)	124 (130)	
Wheat gluten	18 (15)	180 (150)	180 (150)	180 (150)	180 (150)	180 (150)	180 (150)	
Wheat	61 (60)	61 (60)	61 (60)	61 (60)	61 (60)	61 (60)	61 (60)	
Field peas	100	98	95	90	85	80	60	
Fish oil, capelin	35 (44)	35 (44)	35 (44)	35 (44)	35 (44)	35 (44)	35 (44)	
Rapeseed oil	79 (88)	79 (88)	79 (88)	79 (88)	79 (88)	79 (88)	79 (88)	
Linseed oil	22	22	22	22	22	22	22	
Palm kernel oil	44 (48)	44 (48)	44 (48)	44 (48)	44 (48)	44 (48)	44 (48)	
Histidine	0.00	0.34	0.68	1.36	2.04	2.72	5.44	
DL methionine	0.00	0.13	0.25	0.51	0.76	1.02	2.04	
Taurine	0.00	0.61	1.22	2.45	3.67	4.90	9.80	
Minerals*	0.00	0.03	0.06	0.12	0.17	0.23	0.47	
Water soluble vitamins	0.00	0.01	0.03	0.06	0.09	0.12	0.24	
Fat soluble vitamins	0.00	0.03	0.06	0.12	0.17	0.23	0.46	
Cholesterol	0.00	0.28	0.56	1.13	1.69	2.25	4.50	
Notes.

* All diets were added 38 g kg−1 monosodium phosphate, mineral additions were adjusted to each nutrient package. All diets were added 9.3 g kg−1 lysine, 1.8 g kg−1 threonin, 8 g kg−1 choline (50%), 0.25 g kg−1 barox becp dry. Detailed composition for the B-vitamins are given in Table 2.

Table 2 Analysed feed and nutrient composition.

All results are the mean of two analytical parallels. Protein, lipid, starch ash and dry matter are given in g kg−1, while all other diet components are given as mg kg−1. There was a slight difference in macronutrient composition; trial 2 is given in parenthesis. The micronutrients in diets were similar in trial 1 and 2.

	0NP	25NP	50NP	100NP	150NP	200NP	400NP	
Proximate composition, g kg−1	
Protein	453 (480)	469 (472)	449 (440)	456 (480)	462 (480)	470 (480)	461 (480)	
Lipid	213 (220)	203 (220)	219 (210)	211(230)	208 (220)	197 (240)	195 (220)	
Starch	112	112	109	104	106	107	94	
Ash	66	68	66	67	69	60	75	
Dry matter	910 (950)	930 (940)	920 (930)	920 (930)	930	920 (930)	920	
Energy, joule kg−1	22.8	22.7	22.6	22.7	22.4	22.5	22.0	
Micronutrients and other dietary components, mg kg−1	
Vitamin B6	2.6	3.2	4.0	5.2	7.1	10.0	16.0	
Biotin	0.25	0.28	0.31	0.40	0.57	0.70	1.14	
Cobalamin	0.07	0.06	0.10	0.17	0.30	0.39	0.69	
Folate	1.18	1.17	1.58	2.25	3.28	4.20	7.23	
Pantothenic acid	6.5	7.5	9.9	16	25	31	54	
Riboflavin	2.7	3.2	4.7	6.8	10	13	20	
Thiamine	3.4	4.2	5.0	6.2	8.3	11	16	
Niacin	36	36	41	50	64	76	110	
Vitamin C	<5.5	14	28	63	89	140	170	
Methionine g kg−1	6.99	7.01	7.17	7.29	7.86	8.08	8.49	
Histidine g kg−1	10.5	10.4	11.0	11.3	12.3	12.4	13.2	
Taurine g kg−1	0.70	1.20	1.79	3.10	3.90	5.27	5.92	
Lysinea g kg−1	29.7	31.2	28.1	28.4	29.5	28.6	31.0	
Notes.

a As a representative of the amino acids that were not varied between the diets (either added to all diets in equal amounts or only provided naturally by the diet ingredients).

Feeding trials

Both feeding trials were conducted in accordance with Norwegian laws and regulations concerning experiments with live animals, which are overseen by the Norwegian Food Safety Authority. Permission for these specific experiments were given by the Directorate of Fisheries, and accepted for feeding trials at GIFAS, §13 (Akvakulturloven) and §28a (Lakseforskriften) (ref 13/11363), and acknowledged by the advisory board 27.11.12 (ref: ARRAINA regression trial permission).

Trial 1: the trial with parr in freshwater took place at the Institute of Marine Research (Matredal, 61°N, Western Norway). The salmon were hatched in February, and in June the salmon parr were randomly distributed in fifteen 400 litre (1 × 1 × 0.4m) experimental tanks, using a flow-through system with 20 L min−1. The fish was acclimated for one week while being fed commercial feed (Skretting ARC, Norway). The trial commenced on July 3rd, with duplicate tanks for each diet, with the exception of NP100 that was run in triplicates. Each tank contained 100 fish with mean initial body weight of 18.3 ± 2.2 g (mean ± standard deviation (SD)). The fish were fed ad libitum with continuous feeding from automated feeders and a 24h light regime (continuous light). However, care was taken to limit overfeeding due to uncertainties in the collection of uneaten feed at such a small pellet size. Collection and weighing of uneaten feed was conducted daily at 13:00, with the exception of weekends. The fish were exposed to continuous light, and oxygen saturation was monitored on a regular basis and was never below 75%. The fish were reared in freshwater, but with seawater added as a buffer, creating a salinity of 1.1–1.3 g L−1. The temperature was kept constant throughout the experiment, at 12.4 ± 0.7 °C (mean ± SD). The total duration of the feeding trial was 12 weeks.

Trial 2: post-smolt Atlantic salmon were randomly distributed among fifteen sea cages (5m × 5m × 5m; 125 m3; 150 fish per cage) at Gildesskål Research Station, GIFAS, Gildeskål kommune, Norway. Prior to the start of the trial, fish were acclimated to the environmental conditions for two weeks; the feeding trial started in January 2013. As in the freshwater trial, duplicate cages were fed each diet, while the 100NP diet was fed to fish in triplicate cages. At the start, the mean ± SD fish weight was 228 ± 5g, and during the 157 day feeding period the fish more than doubled in weight. As in standard aquacultural practice, fish were reared under 24h light regime before the start of the trial and during the first three months of the experiment. Cages were illuminated by four 400W IDEMA underwater lights that were positioned at the centre of each block of four cages at a depth of three meters. Fish were hand-fed to satiation twice a day and feed intake was recorded for each sea cage. Mortality was recorded daily. Water temperature, salinity and oxygen saturation over the course of the trial varied from 4.1 (January) to 10 °C (June), 30–34.2 g L−1, and 8.7–12.2 mg L−1, respectively.

Sampling

Fish were anesthetized (Benzoak®VET, 0.2 ml/L, ACD Pharmaceuticals, Leknes, Norway) and killed by a blow to the head. In addition to bulk weighing of the total biomass of the fish in trial 1 (freshwater) at each sampling point, body weight and length were measured on individual fish. Initially, 44 fish were sampled before the experiment commenced, thereafter 32 per tank during the experiment (five per tank after three weeks, 22 in the mid sampling after six weeks and five after nine weeks) and 60–68 per tank at the final sampling after 12 weeks of feeding (dwarf males were excluded to avoid any bias by gonad production; these were not followed up by any more analyses; the registered number of dwarf males was in total 11 out of a population of 1,017, with no difference between tanks or diet groups). The number of fish sampled was based on the following: should be representative for the population in the tank, and should provide enough tissue for all planned analyses. To secure the correct amount of feed given according to biomass, a larger number of fish were weighted. All sampling for analyses was performed in the “feeding state” with no starvation before sampling (due to the effect on the GI intestinal morphology, and the focus on amino acids (Espe et al., 2006). At the end of trial 1, there were some dwarf males present in all tanks; these were all excluded from tissue sampling and from growth data (fish suspected of being dwarf males were opened to confirm maturing of gonads). In trial 2 (seawater), 44 individual fish were weighed at each sampling. In both trials, blood was drawn from the caudal vessel (Vena caudalis) by means of a heparinized medical syringe from eight fish per cage, before organs were dissected out and kept as individual samples. Pooled organ samples based on 10 fish for each cage (liver, gills, muscle) were frozen on dry ice and later homogenized, while individual organ samples were flash frozen in liquid nitrogen. Liver weight and gutted weight were recorded (10 fish per cage). Faeces were collected by stripping from the fish used for pooled whole fish samples (10 fish per cage).

Chemical analysis of diets, whole fish and organs

Moisture was measured by drying at 103 °C for 24 h, ash weighed after burning at 540 °C and lipid after extraction with ethyl-acetate in fish tissue, and acid-extraction in fish feed (Lie, 1991). Nitrogen was measured with a nitrogen analyzer (Vario Macro Cube, CN; Elementar Analysensysteme GmbH, Hanau, Germany) according to AOAC official methods of analysis (Sweeney & Rexroad, 1987), and protein calculated as N × 6.25. Starch was measured by enzymatic degradation as described by Hemre et al. (1989). Total amino acids in diets were determined after hydrolysis in 6N HCl at 22 °C using UPLC (ultra performance liquid chromatography) method as described by Andersen et al. (2015). Free amino acids and nitrogen metabolites in plasma and liver were analyzed using the Biochrome with post column derivatization with ninhydrin as described by Espe et al. (2006). The methyl-donor S-adenosyl methionin (SAM) and S-adenosyl-homocystein (SAH) were determined in liver by HPLC as described (Espe et al., 2008). The B-vitamins biotin, niacin, folate, pantothenic acid and cobalamin were all determined by microbiological methods (Feldsine, Abeyta & Andrews, 2002; Maeland et al., 2000). Some of the B-vitamins were determined by HPLC; thiamine (Lynch & Young, 2000), vitamin B6 (Esteva et al., 1998) and riboflavin (Brønstad, Bjerkå& Waagbø, 2002).

Lipogenic enzyme activity in liver

Enzyme activity of liver NADH-isocitrate dehydrogenase (ICDH, EC 1.1.1.42), malic enzyme (ME, EC 1.1.1.40), glucose-6-phosphate dehydrogenase (G6PDH, EC 1.1.1.49) and 6-phosphogluconate dehydrogenase (6PGDH, EC 1.1.1.44) was analysed in fish from trial 1 freshwater. The method is described in detail by Sanden (2001). Analysis of total protein were performed on Maxmat Biomedical Analyser (SM1167; Maxmat S.A., Montpellier, Herault, France) using Maxmat reagents and the appropriate calibrators, standards and controls for the method. The specific lipogenic enzyme activities are expressed as μmol min−1 mg protein −1.

Haematology, plasma enzymes and nutrients

Of the pooled blood samples, 200 μl were kept as individual samples for haematology, while the remainder was centrifuged at 3000 × g for 10 min to obtain the plasma fraction, which was immediately frozen in liquid nitrogen and stored at −80 °C. Hematocrit (Hct) was immediately measured using Vitex Pari microhaematocrit tubes and a Hettich centrifuge (type 201424). Red blood cells (RBC) and haemoglobin (Hb) were measured on a Cell-Dyn 400 (Sequoia-Turner) according to the manufacturer’s instructions, using Para 12 control blood (Streck) for calibration. Plasma ASAT and ALAT (alanine amino transferase), and the indices mean cell volume (MCV), mean cell haemoglobin (MCH) and mean cell haemoglobin concentration (MCHC) were calculated according to Sandnes, Lie & Waagbø (1988). Blood smears were prepared at sampling, following the procedure described by Sandnes, Lie & Waagbø (1988).

Gene expression analysis

Total RNA was purified from frozen liver using the EZ1 RNA Universal Tissue Kit on the BioRobot® EZ1 (Qiagen, Hilden, Germany), including the optional DNase treatment step in the protocol. Homogenisation in QIAzol lysis reagent from the kit was performed on the bead grinder homogeniser Precellys 24 (Bertin Technologies, Montigny-le-Bretonneux, France) for 3 × 10 sec at 6,000rpm. Quantity and quality of RNA were assessed with the NanoDrop® ND-1000 UV-Vis Spectrophotometer (NanoDrop Technologies, Wilmington, DE, USA) and a selection of samples were evaluated on the Agilent 2100 Bioanalyzer (Agilent Technologies, Palo Alto, CA, USA), with the 6000 Nano LabChip® kit (Agilent Technologies, Palo Alto, CA, USA). Average RNA integrity number (RIN) for the samples was 9.4 ± 01 (mean ± SEM, n = 11).

Reverse transcription (RT) was performed on a GeneAmp PCR 9700 (Applied Biosystems) using the TaqMan®reverse transcriptase kit with oligo(dT) primers (Applied Biosystems). Primer sequences are given in Table 3. Samples were run in duplicate (500 ng, ± 5%), in addition to a six point dilution curve in triplicate (1,000 to 31.25 ng), non-template and non-amplification controls. Real-time PCR amplification and analysis were performed on a LightCycler 480 Real-time PCR system (Roche Applied Science, Basel, Switzerland) with SYBR® Green I Mastermix (Roche Applied Science). Thermal cycling was done for 45 cycles of 10 sec at each of 95°C, 60°C and 72°C (basic programme from Roche), followed by melting analysis to confirm that only one product is present.

Table 3 RT-qPCR assays used, including gene names, accession numbers, primer sequences, amplicon sizes and PCR efficiencies.

Gene	Full gene name	Accession/ Contig no.	Forward primer	Reverse primer	Amplicon size (bp)	PCR efficiency	
MCC1	Methylcrotonoyl-CoA carboxylase subunit alpha, mitochondrial precursor	Unigene882_A.salmon	TCCTCCATCATC TGCCATCA	GGAGCTTCTCATCC GATTGG	122	1.92	
G6PD	Glucose-6-phosphate dehydrogenase	Contig7869_A.salmon	GCCGTCTACGCCA AGATGAT	TCGGGCAACTTCAC ATCCTT	110	2.08	
PCCA	Propionyl CoA Carboxylase, Alpha Polypeptide	Unigene11719_A.salmon	GCTGAAGGCCAG GAGATCTG	TCCAACCGTCTCT CCTGCTT	114	2.16	
PCCB	Propionyl-CoA carboxylase beta chain, mitochondrial-like	Contig4808_A._salmon	CCGTCCCCAAGAT CACCAT	CAGAAGGCCAGGC GTAGTTC	105	2.16	
ACTBa	Beta-actin	BG933897	CCAAAGCCAACAGG GAGAA	AGGGACAACACTGC CTGGAT	92	2.00	
EEF1ABa	Eukaryotic translation elongation factor 1 alpha B	BG933853	TGCCCCTCCAGG ATGTCTAC	CACGGCCCACAG GTACTG	57	2.03	
UBA52a	Ubiquitin A-52 Residue Ribosomal Protein Fusion Product 1	NM_001141291	CCAATGTACAGCGC CTGAAA	CGTGGCCATCTTGA GTTCCT	110	2.04	
Notes.

a reference genes.

Cycle threshold (Ct)-values were calculated using the second maximum derivative method in the Lightcycler® software. Normalization to the reference genes (ACTB, EEF1A and UBA52) and data analysis was conducted with the geNorm applet (Vandesomplele et al., 2002).

Calculations

Condition factor (K) = (bw/fl3)*100 (bw = body weight, fl = fork length)

Organosomatic index (%) = (ow/bw)*100 (ow = organ weight)

Specific growth rate (%/day) = (lnWf − lnWi*100)/t (Wf = final weight, Wi = initial weight, t = time in days)

Thermal growth coefficient = 1000*(Wi ˆ1/3 − Wf ˆ1/3)/ T*t (T = temperature) (%/day adjusted to water temperature)

MCV = (Hct/rbc)*10

MCH = (Hb/rbc)*10

MCHC = (Hb/Hct)*100

Apparent digestibility coefficient, % = 100 − ((%yttriumin feed/ %yttrium in faeces)*(%nutrient in faeces/ % nutrient in feed)*100))

Nutrient retention, % = [(final biomass × final nutrient content) − (initial biomass × initial nutrient content)] × 100/ (Total feed intake × nutrient content in feed).

Statistical analyses

All data are given as mean for each diet group and the pooled standard deviation, as most analyses were conducted on pooled samples (n = 2 or n = 3 tanks for each dietary group), thus separate standard deviations for each diet group would not be very informative. Regression analyses were conducted in Statistica (Version 11; Statsoft, Tulsa, OK, USA), using tank values (either from pooled samples or tank means for parameters where individual data were available). For each response parameter, an attempt was made to find the best fit model for the data; model comparison was conducted using GraphPad Prism 6. In some cases, the second order polynomial regression fit the data significantly better than the first order polynomial (straight line). Due to the multi-approach with all nutrients added simultaneously, it was not possible to run regression with analyzed feed level as independent factor for many of the response parameters (such as growth, organ indices, hematology, most amino acids etc.). Consequently, the levels 0, 25, 50, 100, 150, 200 and 400 were used. For the nutrients which were fitted with a second order polynomial, the model was used to calculate the feed level at which maximum tissue status would be achieved, thus analyzed feed concentrations rather than 0, 25, 50 etc were used for these. In these five cases, the deviation in adjusted R2-values between analysis conducted with actual feed values and 0, 25, 50 etc were 3 percentage points on average and it varied which method gave the higher values. Based on this, it was concluded to stick to using 0, 25, 50 etc for the other nutrients, as this would make little difference, and the linear regressions could not be used for calculation of recommended feed concentrations of the different nutrients anyway. For regression models, adjusted R2 and p-values are given when the latter is <0.05. For the parameters where data on individual fish were available (weight, length, HSI (hepatosomatic index)), (VSI (viscerasomatic index) and K), a nested ANOVA was conducted with diet as fixed factor and tank as random factor, to account for individuals from the same tank being pseudoreplicates.

Table 4 Performance parameters and organ weights trial 1 with parr in freshwater.

Mean and pooled SD. Regression analysis is conducted on tank means (n = 2 per diet group, and n = 3 for 100NP).

	0NP	25NP	50NP	100NP	150NP	200NP	400NP	Pooled SD	Regression	
SGR	1.75	1.75	1.74	1.75	1.82	1.82	1.86	0.04	R2= 0.48, p = 0.003	
TGC, total	1.60	1.60	1.58	1.60	1.69	1.68	1.73	0.04	R2= 0.48, p = 0.003	
Final weight	79.7	79.6	78.6	79.7	84.6	84.2	87.3	2.5	R2= 0.49, p = 0.002	
Length	19.5	19.5	18.2	18.4	18.7	18.6	18.7	0.7	n.s.	
K	1.28AB	1.26A	1.28AB	1.28AB	1.29AB	1.30B	1.34C	0.01	R2= 0.70, p < 0.0001	
FCR	0.86	0.83	0.82	0.83	0.85	0.78	0.81	0.02	n.s.	
HSI	1.53A	1.60A	1.56A	1.65A	1.31B	1.25B	1.25B	0.07B	R2= 0.46, p = 0.03	
VSI	11.6AB	11.9A	11.9A	11.9A	10.8BC	10.6C	10.5C	0.31	R2= 0.48, p = 0.02	
Notes.

Initial weight 18.3 (2.2) g, length 11.4 (0.5) cm, K 1.23 (0.04), HSI 1.20 (0.32), VSI 14.8 (1.5), all as mean ± sd for 44 fish (15 for the organ indices).

K condition factor

HSI hepatosomatic index

VSI viscerosomatic index

Nested ANOVA was conducted for the parameters where individual data was available.

Results

Diets

All diets contained similar amounts of protein, lipid, energy, ash and dry matter (Table 2). There seemed to be a slight gradient in starch content, probably due to the NP replacing field peas in the diet formulation. The added nutrient package generally exhibited a dose–response design, while concentrations of compounds that had not been part of the NP, such as tocopherols other than α-tocopherol, vitamin A2, lysine and phytosterols were similar in all seven feeds. Apart from the added nitrogen compounds, amino acid profiles were similar in all feeds (Espe et al., 2014). All diets contained the same oil mixture and fatty acid composition. The marine n-3 fatty acids EPA and DHA constituted 2.5% of the dietary fatty acids in both trials, but due to increasing dietary lipid content with increasing fish size, this amounted to 3.9 g kg −1 feed in trial 1 and 4.8 g kg−1 in trial 2 (further data on fatty acid composition are not presented).

Fish performance

In trial 1, fish grew from an initial weight of 18.3 g (±2.2) to a range of 78.6 g (±1.9) to 87.3 g (±4.5). Both fish growth and protein retention increased with increasing dietary NP, while lipid retention decreased, together with liver index and viscera-somatic index (Table 4; p < 0.05). In trial 2, initial size was 228 g (±4.2), and average final weight was 482 g (±17). In the seawater trial with post-smolt, there was no effect of the nutrient package on growth or protein and lipid retention (Table 5). Survival was high in both trials, close to 100%, and with no difference between diet groups.

Table 5 Performance parameters and organ weights trial 2 with post-smolt in seawater.

Mean and pooled SD. Regression analysis is conducted on means for each cage (n = 2 per diet group, and n = 3 for 100NP).

	0NP	25NP	50NP	100NP	150NP	200NP	400NP	Pooled SD	Regression	
SGR	0.47	0.50	0.48	0.50	0.50	0.48	0.50	0.02	n.s.	
TGC	2.37	2.53	2.42	2.52	2.53	2.46	2.55	0.10	n.s.	
Initial weight	224	229	230	227	229	228	231	4.2	n.s.	
Final weight	459	487	480	480	492	478	496	17	n.s.	
Length	34	34	34	35	36	34	37	2		
K	1.21	1.36	1.19	1.09	1.01	1.18	1.00	0.2	n.s.	
FCR	0.99	0.98	0.99	0.97	0.97	0.99	0.97	0.02	n.s.	
HSI	7.80	7.99	7.86	8.18	7.81	7.59	7.84	0.41	n.s.	
VSI	1.34	1.42	1.37	1.33	1.46	1.31	1.32	0.07	n.s.	
Notes.

P1 period 1, week 0 to 6

P2 period 2, week 6 to 12

K condition factor

HSI hepatosomatic index

VSI viscerosomatic index

Nested ANOVA was conducted for the parameters where individual data was available.

Table 6 Haematology of fish fed the experimental diets during the parr freshwater stage (trial 1), with one pooled sample of blood/plasma from eight fish analysed per tank at the final sampling.

	0NP	25NP	50NP	100NP	150NP	200NP	400NP	Pooled SD	Regression	
Hb	9.05	9.15	8.80	9.13	9.25	9.15	9.40	0.15	R2= 0.25, p = 0.03	
RBC	1.27	1.32	1.30	1.31	1.27	1.28	1.29	0.04	n.s.	
Hct	42.5	42.5	41.5	41.7	43.0	42.0	43.0	0.6	n.s.	
MCV	335	322	319	319	340	330	335	9	n.s.	
MCH	71.3	69.3	67.7	70.0	73.3	71.9	73.2	1.9	n.s.	
MCHC	21.3	21.5	21.2	21.9	21.5	21.8	21.9	0.4	n.s.	
Notes.

Initial values at the start sampling: Hct 38.0 (1.7), RBC 1.19 (0.07), Hb 8.13 (0.31), MCV 319 (8), MCH 68.3 (3.4), MCHC 21.4 (1.1) , all as mean (sd) of three analyzed samples, each sample pooled from 5 fish

RBC red blood cell count

Hct hematocrit

MCV mean cell volume

MCH mean cell haemoglobin

MCHC mean cell haemoglobin concentration

In trial 1, linear regression analysis showed that fish growth significantly increased by increasing dietary NP inclusion (R2 = 0.48, p = 0.003) (Table 4). Final weight (R2 = 0.49, p = 0.002) and condition factor (K, R2 = 0.70, p < 0.0001) also increased with increasing NP addition, while length was not affected. Nested ANOVA showed differences between individual diet groups in condition factor (K); fish fed 400NP had higher K than all other diet groups, while fish fed diet 200NP had significantly higher K than fish fed 25NP. For final weight, individual variation was too large to produce significant differences by ANOVA (p = 0.13). Both hepatosomatic index (HSI, R2 = 0.46, p = 0.03) and viscerosomatic index (VSI, R2 = 0.48, p = 0.02) decreased with increasing NP addition. Nested ANOVA showed that HSI was significantly higher in diet group 0NP-100NP, compared to 150NP-400NP (p = 0.005). The same was the case for VSI (p = 0.01), except 150NP did not differ significantly from 0NP. In trial 1 there was a pattern for HSI, VSI, SGR and final weight where the four diets with the lowest NP (0–100NP) had similar values to each other, while the three diets with the most NP (150NP–400NP) also exhibited similar values to each other, with lower organosomatic indices and better growth in the latter three groups. Feed conversion ratios were not significantly affected by diet. Post-smolt held in seawater showed equal SGR and FCR, HSI, VSI and K, so for the larger fish no diet effect during doubling of weight was observed.

Table 7 Heamatology of fish fed the experimental diets during the post-smolt seawater stage (trial 2), with one pooled sample of blood/plasma from 8 fish analyzed cage at the main sampling.

Abbreviations, see Table 6.

	0NP	25NP	50NP	100NP	150NP	200NP	400NP	Pooled SD	Regression	
Hb	9.4	9.0	9.2	8.9	9.1	9.1	9.6	0.4	n.s.	
RBC	1.35	1.30	1.29	1.32	1.31	1.29	1.35	0.03	n.s.	
Hct	39	37	38	38	38	38	40	1.6	R2= 0.2 p = 0.041	
MCV	289	281	291	286	290	294	299	9	R2= 0.3 p = 0.043	
MCH	69	69	71	68	70	70	71	2	n.s.	
MCHC	24	25	24	24	24	24	24	1	n.s.-	

Table 8 Proximate analyses and nutrient retention from trial 1 with parr in freshwater, homogenized pooled whole fish samples of 10 fish per tank, (n = 2 per diet group, and n = 3 for 100NP).

Data are presented as mean ± pooled sd, proximate analyses in g kg−1.

	0NP	25NP	50NP	100NP	150NP	200NP	400NP	Pooled SD	Regression	
Proximate analyses (g kg−1)	
Lipid	132	129	131	129	127	117	119	2.7	R2= 0.53, p = 0.001	
Protein	175	180	175	180	175	185	190	3.9	R2= 0.43, p = 0.005	
Ash	23.3	23.6	21.2	24.5	19.8	23.3	22.8	2.2	n.s.	
Dry matter	330	330	330	330	320	325	320	2.0	R2= 0.52, p = 0.001	
Nutrient retention (%)	
Protein	45.4	47.1	48.2	48.8	45.0	52.3	52.5	2.1	R2= 0.31, p = 0.02	
Lipid	78.0	82.0	78.5	79.9	76.9	79.6	78.6	3.4	n.s.	
Notes.

Initial levels before the start of the trial (analyzed on triplicate samples, each consisting of five pooled fish), mean with SD in parenthesis: lipid 102 (3), protein 170 (0), ash 22.1 (0.5), dry matter 285 (1).

Haematology and blood smears

In trial 1, the only plasma parameter that was significantly affected by diet was ASAT (R2 = 0.83, p < 0.0001), with higher values when NP inclusion increased (Table 6). The only haematological parameter affected by diet in Trial 1 with small fish was haemoglobin (Hb), increasing with higher NP (R2 = 0.25, p = 0.03, Table 6). In larger salmon (trial 2) hematocrit (Hct, p = 0.041 R2 = 0.2) and mean cell volume (MCV) responded with increased values as NP increased (p = 0.043, R2 = 0.3) (Table 7).

Blood smears from all groups were investigated and no groups showed any sign of variance between blood cells, or any sign of megaloblastic anemia. The blood smears were visually inspected by means of a microscope (40x enlargement).

Table 9 Proximate analyses and nutrient retention from trial 2 with postsmolt in seawater, homogenized pooled whole fish samples of 10 fish per cage, (n = 2 per diet group, and n = 3 for 100NP).

Data are presented as mean ± pooled sd, proximate analyses in g kg−1.

	0NP	25NP	50NP	100NP	150NP	200NP	400NP	Pooled SD	Regression	
Proximate analyses (g kg−1)	
Lipid	89	97	95	96	90	83	96	17	n.s.	
Protein	180	195	190	190	195	195	190	9.6	n.s.	
Ash	20	18	20	19	19	20	21	1.7	n.s.	
Dry matter	293	301	299	296	296	288	297	4.9	n.s.	
Nutrient retention (%)	
Protein	33.1	41.0	41.1	38.4	40.6	39.7	38.4	4.9	n.s.	
Lipid	43.2	51.0	51.6	48.2	45.2	33.9	50.0	6.6	n.s.	
Notes.

Initial levels before the start of the trial (analyzed on triplicate samples, each consisting of five pooled fish), mean with SD in parenthesis: lipid 85 g kg−1, protein 200 g kg−1.

Table 10 Free amino acids and N-metabolites in plasma (μmol/dl) and liver (μmol/g) of Atlantic salmon fed the seven experimental diets, analyzed on pooled samples of eight fish per tank for plasma and 10 fish per tank for liver, 2–3 tanks per diet group.

Values are given as diet group means, and the pooled standard deviation is given. Results are from parr in freshwater (Trial 1).

	0NP	25NP	50NP	100NP	150NP	200NP	400NP	Pooled SD	Regression	
Free amino acids and N-metabolites in plasma (μmol/dl)	
TAU	61.8	49.9	56.7	71.8	77.5	82.4	88.2	3.9	R2= 0.66, p = 0.0001	
GLY	26.3	23.5	24.3	24.8	23.7	18.7	18.7	1.9	R2= 0.46, p = 0.003	
VAL	54.7	53.4	48.9	45.2	46.4	43.9	43.7	4.1	R2= 0.25, p = 0.03	
HIS	10.9	8.8	8.7	9.0	9.3	10.3	11.1	0.9	n.s. (p = 0.13)	
ARG	12.2	9.4	11.4	11.4	13.5	13.4	13.2	1.3	n.s. (p = 0.08)	
Free amino acids and N-metabolites in liver (μmol/g)	
TAU	15.7	16.0	16.8	18.1	17.4	15.3	17.9	1.06	n.s.	
PEA	0.22	0.23	0.23	0.22	0.21	0.19	0.19	0.01	R2= 0.42, p = 0.008	
OH-PRO	0.15	0.16	0.11	0.09	0.09	0.09	0.09	0.02	R2= 0.27, p = 0.03	
THR	0.87	1.04	0.93	0.86	0.79	0.81	0.73	0.07	R2= 0.45, p = 0.005	
SER	1.59	1.86	1.67	1.44	1.40	1.42	1.27	0.10	R2= 0.47, p = 0.004	
GLU	7.38	7.41	7.67	7.63	8.11	8.22	9.04	0.36	R2= 0.65, p = 0.0003	
GLN	4.53	4.79	4.80	4.10	4.35	3.91	3.91	0.32	R2= 0.22, p = 0.05	
PRO	1.26	1.50	1.26	1.17	1.16	1.14	1.06	0.08	R2= 0.34, p = 0.02	
MET	0.34	0.37	0.33	0.33	0.35	0.42	0.42	0.03	R2= 0.26, p = 0.04	
HIS	0.76	0.78	0.78	0.71	0.75	0.72	0.80	0.04	n.s.	
Liver methyl donors (nmol/g ww)	
SAH	24.4	23.5	25.2	21.9	19.6	26.3	21.0	2.4	n.s.	
SAM	40.3	30.1	35.6	33.2	37.2	40.3	40.2	4.1	n.s.	
Notes.

Tau taurine

gly glycine

val valine

his histidine

arg arginine

pea phosphoethanolamine

OH-pro hydroxyl-proline

thr threonine

ser serine

glu glutamine

gln glutamic acid

pro proline

met methionine

HIS one-methylhistidine

SAH S-Adenyhomocysteine

SAM S-adenosyl-methionine

Table 11 Free amino acids and N-metabolites in plasma (μmol/dl) and liver (μmol/g) of Atlantic salmon fed the seven experimental diets, analyzed on pooled samples of eight fish per tank for plasma and 10 fish per tank for liver, 2–3 tanks per diet group.

Values are given as diet group means, and the pooled standard deviation is given. Results are from smolt in seawater (Trial 2).

	0NP	25NP	50NP	100NP	150NP	200NP	400NP	Pooled SD	Regression	
Free amino acids (μmol/g)	
TAU	3.3	6.4	11.5	49.2	57.3	47.2	57.9	3.3	R2= 0.51, p = 0.0016	
THR	18.4	17.0	17.0	15.2	14.5	12.6	15.2	0.8	R2= −0.21, p = 0.048	
SER	25.1	24.9	22.6	19.5	16.7	14.0	14.3	0.5	R2= −0.71, p < 0.001	
GLN	53.9	50.7	40.3	39.5	38.8	28.0	34.9	1.7	R2=−0.41, p = 0.006	
PRO	88.9	84.6	84.3	61.4	63.4	46.4	49.3	8.3	R2= −0.46, p = 0.003	
GLY	94.3	84.6	86.2	68.9	66.4	50.6	50.4	3.0	R2= −0.71, p = 0.001	
Citrulline	6.4	6.4	7.0	11.7	101.	13.5	17.7	2.2	R2= 0.48, p = 0.002	
AABA	0.8	0.7	0.7	0.9	1.6	1.2	1.4	0.2	R2= 023, p = 0.04	
VAL	59.9	56.7	51.7	48.5	52.9	38.8	43.5	2.0	R2= −0.44, p = 0.004	
LEU	53.0	47.5	47.3	48.6	49.4	38.0	42.7	2.3	R2= −0.26, p = 0.03	
TYR	29.9	27.0	20.6	21.1	14.3	14.5	14.2	2.8	R2= −0.45, p = 0.004	
PHE	18.2	16.9	15.9	18.0	15.8	14.3	14.8	0.8	R2= −0.24, p = 0.04	
Ornitine	4.7	3.6	3.4	2.2	3.1	2.0	2.0	0.3	R2= 0.35, p = 0.011	
ARG	13.5	11.4	12.1	10.0	12.0	8.9	9.2	1.1	R2= −0.25, p = 0.03	
Liver methyl donors (μmol/g)	
SAH	25.7	24.2	26.4	18.0	17.9	18.2	18.9	4.4	n.s.	
SAM	56.0	38.4	52.3	23.9	19.7	24.1	17.7	16.4	n.s.	
Notes.

Tau taurine

thr threonine

ser serine

gln glutamic acid

pro proline

gly glycine

beta-ala bala-beta-alanine

aaba alpha aminobuturic acid

val valine

leu leucine

tyr tyroxine

phe phenylanalnine

arg arginine

SAH S-Adenyhomocysteine

SAM S-adenosyl-methionine

Nutrient retention

Salmon parr held in freshwater responded to increasing NP inclusion with an increase in whole body nitrogen content (R2 = 0.48, p = 0.003, Table 8) and a decrease in whole body lipid content (R2 = 0.53, p = 0.001), the latter was also reflected in a decrease in dry matter (R2 = 0.52, p = 0.001). Protein retention increased with increasing NP (R2 = 0.31, p = 0.02, Table 8). Larger fish held in seawater (trial 2), had similar body composition independent of dietary NP additions (Table 9).

Nutrient status in fish tissues and plasma

There was no feed deprivation before sampling in any of the tanks / cages. Plasma taurine (R2 = 0.66, p = 0.0001, Table 10) and methionine (R2 = 0.68, p < 0.0001) increased with increasing NP addition, glycine and valine, on the other hand, decreased. In muscle, free methionine increased (R2 = 0.26, p = 0.04) together with glutamic acid (R2 = 0.65, p = 0.003). The other free amino acids and N metabolites were unaffected by dietary treatment. Although a linear regression model was used for plasma concentrations of the added methionine, histidine and taurine, all of these had the same pattern when data were inspected visually, where they were relatively high in the 0NP diet, then decreased, before increasing again with increasing NP addition in the diet (Table 10; trial 1, and 10 trial 2). No effect of diet was seen on SAH and SAM in liver (Tables 10 and 11).

Levels of B-vitamins in fish tissues are given in Table 12 (trial 1) and Fig. 1 (B-vitamins in muscle), 2 (B-vitamins in liver), 3 (B-vitamins in whole body) and 4 (panthotenic acid in gills) for trial 2. Muscle thiamine level in trial 1 and trial 2 was close to similar in all groups (around 0.7 mg/g wet weight), but with a weak regression dependent on NP diet level (p = 0.03 trial 1 and p = 0.05 trial 2). Expression of the thiamine dependent G6PD was substantially higher in group fed the 0NP diet compared to the other groups (p = 0.025), while regression analyses showed a weak effect of diet on enzyme activity (p = 0.025) (only measured in trial 1). Muscle tissue thiamine status was also correlated to dietary NP in trial 2 (R2 = 0.45, p = 0.016 trial 2). Riboflavin showed high correlation to dietary NP (R = 0.85, p = 0.00 in trial 1, but with no significant differences in trial 2; Fig. 1). Cobalamin in muscle (R2 = 0.70, p = 0.003 in trial 1, not measured in trial 2), and in liver (R2 = 0.43, p = 0.02, and R2 = 0.53, p = 0.002 in trial 2) responded significantly to dietary B12 in both trials. Liver tissue folate (R2 = 0.47, p = 0.003 in trial 1, and R2 = 0.42 p < 0.01 in trial 2) increased significantly with the increased NP in diet in both trials. Significant linear regressions were seen in whole fish for niacin (R2 = 0.92, p < 0.0001) in trial 1, and trial 2. A second order polynomial model was a significantly better fit to the data than linear regression for pyridoxine in whole fish (R2 = 0.93, p < 0.0001 in trial 1, and R2 = 0.7, p < 0.01 in trial 2). Biotin in whole fish showed similar levels with no difference in concentrations dependent on diet level, neither in small nor larger salmon. Pantothenic acid in gill soft tissue (R2 = 0.95, p < 0.0001 in trial 1, and R2 = 0.95, p < 0.001 in trial 2) responded significantly to diet levels, in both trials. For these vitamins, the models were used to calculate the feed concentration required to obtain maximum tissue levels, as shown in Figs. 1–4.

Table 12 Tissue vitamin status trial 1 (mg kg−1 wet weight), analyzed on homogenized pooled samples of 10 fish per tank, (n = 2 per diet group, and n = 3 for 100NP). Data are presented as mean ± pooled sd.

The column called “Regression” gives R2 and p-values for linear (1st order polynomial) regressions, or for second order polynomial regressions where this provided a significantly better fit to the data. Trial 1 in freshwater.

	0NP	25NP	50NP	100NP	150NP	200NP	400NP	Pooled SD	Regression	
Whole fish	
Biotin	0.081	0.076	0.078	0.076	0.075	0.080	0.079	0.003	n.s.	
Niacin	24.0	24.5	27.0	31.0	36.5	46.0	54.5	1.7	R2= 0.92, p < 0.0001	
Muscle tissue	
Thiamine	0.70	0.65	0.65	0.77	0.75	0.70	0.80	0.4	R2= 0.27, p = 0.03	
B6	1.65	1.75	2.25	3.70	4.30	5.25	6.15	0.39	R2= 0.92, p < 0.0001	
B12	0.033	0.020	0.033	0.045	0.047	0.049	0.052	0.004	R2= 0.63, p = 0.003	
Riboflavin	0.85	0.8	1.15	1.17	1.05	1.25	1.00	0.06	R2= 0.85, p = 0.000	
Liver tissue	
Folate	10.8	9.6	11.1	11.6	10.5	13.5	12.9	0.5	R2= 0.46, p = 0.005	
B12	0.40	0.41	0.41	0.47	0.45	0.48	0.45	0.02	R2= 0.34, p = 0.02	
Gill soft tissue	
B5	2.60	2.30	2.95	4.07	5.50	5.25	5.90	0.19	R2= 0.94, p < 0.0001	

Specific biomarker results

Muscle ASAT was only measured in trial 1, where activated muscle ASAT (with pyridoxine-5-phosphate; P5P) did not respond to variations in dietary inclusion of premix, while the ASAT (not activated) highly correlated with premix inclusion, but with an indication of a plateau at a pyridoxine inclusion level between 8 and 12 mg kg−1 diet (Table 12, Fig. 5; trial 1). Liver mRNA expressions of the gene MCC1 (methyl crotonyl CoA carboxylase; biotin dependent) did not show any difference between diet groups (Table 13). The two biotin-dependent genes PCCA and PCCB (propioyl CoA carboxylase A and B) showed no response for the A form, and a slight but significant regression for the B form. Glucose-6-phosphate dehydrogenase (G6PDH; thiamine dependent) showed high levels in the 0NP group, and then decreased to a more or less stable expression for the 100NP to 400NP diets (Table 13). For these analyses, only trial 1 samples were evaluated. In trial 1 the activity level of enzymes in the pentose-phosphate shunt, 6PGDH (6-phosho-gluconate dehydrogenase, 1.1.1.49; R2 = 0.41, P = 0.01), and G6PDH (glucose-6-phosphate dehydrogenase, 1.1.1.44; R2 = 0.25, p = 0.06) decreased with increasing NP. A similar trend was seen for ME (malic enzyme, EC 1.1.1.39; R2 = 0.21, p = 0.09), while ICDH (iso-citrate-dehydrogenase, EC 1.1.1.42) showed stable values independent of dietary NP (Table 14). These enzyme activities were not measured in trial 2 with post-smolt.

Figure 1 (A) B6, pyridoxine in muscle (mg kg−1 on y-axis): p = 0.007, R2 = 0.92; second order polynomial. (B) B12, cobalamin in muscle (mg kg−1 on y-axis), dietary B12 on x-axis; p < 0.01, R2 = 0.7, second order polynomial. (C) thiamine in muscle (mg kg−1 on y-axis), and dietary thiamine (x-axis); p = 0.05, R2 = 0.27: linear regression. (D) riboflavin in muscle (mg kg−1 on y-axis), and dietary riboflavin (x-axis); not significant.

Figure 2 (A) folate in liver (y-axis, mg kg−1), and in diet (x-axis); p < 0.01, R2=0.42; linear regression. (B) B12 (cobalamin) in liver (y-axis, mg kg−1), and dietary B12 (x-axis); p < 0.01, R2 = 0.75, second order polynomial.

Figure 3 (A) niacin in whole body (y-axis, mg kg−1), and xaxis levels in feed; p = 0.001, R2 = 0.95: linear regression. (B) biotin in whole body (y-axis, mg kg−1), and diet levels (x-axis); not significant.

Figure 4 Pantothenic acid in gill tissue (y-axis, mg kg−1), and x-axis with levels in feed; p < 0.001, R2= 0.95; second order polynomial.

Figure 5 Activity of asparatate amino transferase (ASAT) in muscle of salmon fed diets with increasing levels of the nutrient package (defined in Table 2).

The enzyme was activated or not by adding pyridoxine-5-phosphate in the assay buffer. The points for not activated ASAT were fitted to a first- and second order polynomial and the first order polynomial was preferred (p = 0.14, p = 0.01; R2 = 0.85).

Discussion

In salmon parr, lower growth and increased K, HSI and VSI in the 0NP-100NP groups, indicate that these diets were deficient in one or several nutrients. Post-smolt Atlantic salmon showed only a weak increased growth when the NP level was above 150NP. The difference between the two trials may have multiple explanations, e.g., parr generally grow faster than post-smolt, further post-smolt showed initial larger nutrient stores, and also water temperature was lower during the post-smolt study compared to the parr study (Espe et al., 2010; Waagbø, 2010; Torstensen, Espe & Stubhaug, 2011; NRC, 2011).

The interpretation of results from our studies must be taken with care, due to the multi-approach with all nutrients added simultaneously, there may exist confounding factors of which we do not know. Although the increase in dietary methionine was low, from 7.0–8.5 g kg−1 diet in both trials, methionine below the 150NP (<7.9 g kg −1) might have contributed to lower growth and increased HSI, VSI and K, in salmon parr, in agreement with Espe et al. (2014). Post-smolt given low dietary methionine has been found to increase lipid deposition without affecting somatic growth (Espe et al., 2008; Espe et al., 2016); however, in our post-smolt study (trial 2), none of these parameters were affected. In earlier studies, a close interaction between methionine and taurine has been identified in parr when given high plant diets added graded levels of methionine and/or taurine, where methionine highly affected taurine liver concentration, in line with Espe et al. (2008). Taurine was identified to affect sulphur metabolism, and to be an efficient antioxidant protecting the liver cells against apoptosis (Espe & Holen, 2013).

Table 13 Liver mRNA expression of genes dependent on B-vitamins as co-factors in metabolism; only selected groups were analysed.

MCC1 (Methyl crotonyl coA carboxylase: biotin dependent), PCCA (propioyl CoA carboxylase-A and propionyl CoA carboxylase-B; biotin dependent), G6PD Glucose-6-phosphate dehydrogenase (thiamine dependent). Only trial 1 was analysed.

	0NP	100NP	200NP	400NP	Pooled SD	Regression	
MCC1	0.412	0.422	0.396	0.398	0.116	n.s.	
G6PD	0.706	0.246	0.394	0.311	0.264	R2= 0.14. p = 0.025	
PCCA	0.482	0.369	0.459	0.397	0.125	n.s.	
PCCB	0.423	0.381	0.424	0.308	0.101	R2= 0.15. p = 0.017	

Table 14 Liver 6-phosphogluconate dehydrogeanse (6PGDH), glucose-6-phosphate dehydrogeanse (G6PDH), malic enzyme (ME) and NADP-isocitrate dehydrogeanse (ICDH) activity (μmol min−1 mg protein−1) from trial 1 with parr.

With one pooled sample of liver from eight fish analysed per cage. Data are presented as mean ± pooled sd (n = 2 per diet group, and n = 3 for 100NP). Regression analysis is conducted on means for each cage. Specific enzyme activity given as μmol min−1 mg protein−1

	0NP	25NP	50NP	100NP	150NP	200NP	400NP	Pooled SD	Regression	
6PGDH	0.114	0.116	0.119	0.108	0.108	0.108	0.081	0.017	R2= 0.41, p = 0.01	
G6PDH	0.260	0.180	0.166	0.206	0.141	0.137	0.112	0.074	R2= 0.25, p = 0.06	
ME	0.333	0.331	0.356	0.330	0.325	0.403	0.380	0.039	R2= 0.21, p = 0.09	
ICDH	0.270	0.320	0.274	0.284	0.301	0.320	0.274	0.036	ns	

Blood health parameters were within reported normal ranges in both experiments, indicating acceptable fish health in all groups (Sandnes, Lie & Waagbø, 1988). This was further confirmed also by no incidence of cataract, neither in fresh- nor in seawater, and negliable mortality in both dietary trials. Dietary histidine varied from 10.3 to 13.1 mg g−1, the highest level was expected to significantly protect against cataract development, in accordance with Waagbø (2010) and Remoe et al., (2014). In the present study, the lowest dietary histidine did not result in any cataract development, maybe due to limited time on the experimental diets. Higher histidin levels, around 12–14 mg kg−1, are recommended to reduce risk of cataract development in sea (Waagbø, 2010).

In addition to growth, HSI and VSI, B-vitamins requirement can be defined based on body and/or organ concentration saturation together with specific biomarkers for each of the vitamins (Hansen, Waagbø & Hemre, 2015). Dietary vitamin requirement to maintain whole body/tissue saturation is most often higher than the requirement for maximum growth, especially when evaluated during short feeding trials, as also indicated in the present two experiments (Hansen, Waagbø & Hemre, 2015).

The high gene expression of G6PD in the 0NP group might indicate a thiamine dependent metabolic change in this group, since G6PD gene expression may also vary in response to other metabolic adjustments, for example redox regulation, as described in K Hamre et al. (2016, unpublished data). Enzyme activity of G6PDH further confirmed a change in the pentose phosphate activity dependent on NP dose in the diet. This trend was even stronger for the rate-limiting 6PGDH. Both these enzymes participate in the regeneration of NADP+ to NADPH together with ME and ICDH; however no variation was found for the two latter. As a first precaution, we recommend to add some thiamine to plant based diets; at least above 6.2 mg kg−1 as found in the 25NP group. Deficiency signs previously reported for thiamine deficiency, like abnormal swimming, were not observed, as would be expected if thiamine were too low or were broken down e.g., by diet ingredients containing thiaminase, but that would demand raw materials that were of fish origin and which were not subjected to heat (Hansen, Waagbø & Hemre, 2015). With no abnormal swimming observed, the weak regression above the 25NP level we conclude is a response to the dose–response design.

Riboflavin deficiency is reported to cause cataract, photophobia and result in short body vertebrae in fish (Waagbø, 2010), none of these deficiency signs were observed in the present study. However, muscle riboflavin increased dependent on dietary riboflavin, up to a diet level around 5 mg kg−1 diet, after which it flattened, indicating that this level is close to fulfilling all metabolic needs for riboflavin, but only in parr. Fish size can influence requirement, which may explain the lack of response in post-smolt. To secure significant riboflavin, we recommend that plant based diets for Atlantic salmon should be added 5mg kg−1 (50NP), especially in diets for parr.

The requirement for vitamin B6 (pyridoxine, pyridoxal and pyridoxamine and their phosphate esters) for growth has been estimated to vary between 2 and 16 mg kg −1 for different fish species (Hansen, Waagbø & Hemre, 2015). In the present study, no breakpoint was found in muscle B6 level, indicating that as high as 10 mg kg−1 might be favorable. This is higher than the 6–8 mg kg−1 diet, based on saturated muscle ASAT activity, as indicated by Albrektsen, Waagbø & Sandnes (1993). In the present study, muscle ASAT was saturated at a dietary B6 level around 10 mg kg−1 diet, both in salmon parr and post-smolt. There is a close relationship between vitamin B6 and protein metabolism shown in studies with Jian carp (Cyprinus carpio var. Jian) by He et al. (2009). They found improved protein productive value (PPV) and reduced plasma ammonia in fish fed a diet supplemented with vitamin B6 compared to control fish. In our study with parr, PPV was improved at a NP inclusion level of 150NP or higher. Symptoms and consequences of vitamin B6 deficiency are many, diverse and severe, and include nervous disorders and abnormal behavior (Hansen, Waagbø & Hemre, 2015). No behavior disorders were observed in the present study, but to secure growth and optimal protein retention, and saturation of muscle ASAT activity, plant based diets to parr and post-smolt are recommended to contain around 10 mg kg−1 diet B6 or above.

Cobalamin, vitamin B12, reached a maximum liver concentration at a diet concentration of 0.17mg kg−1 similar to what is recommended in NRC (2011). A sign of deficiency would be megaloblastic anemia, characterized by large erythrocytes and white blood cells with fragmented cell nucleus. Often there is a close link to hemoglobin (Hb) when megaloblastic anemia occurs. Hb levels in the present study were all within a normal range for Atlantic salmon (Sandnes, Lie & Waagbø, 1988; Woodward, 1994), although the Hb of parr in freshwater fed the 0NP diet was lower than in the other groups. Blood smears showed that there were no signs of megaloblastic anemia in any of the diet groups. Gut microbiota in warm water fish produce some vitamin B12 (Limsuwan & Lovell, 1981), but even if this took place in the present study (not investigated), salmon diets should still be added vitamin B12 in the premix at a level of 0.17 mg kg−1, or higher to saturate body levels, both for parr and post-smolt. No adverse effects were identified at an addition as high as 0.72 mg kg−1. The main function of vitamin B12 is when methylcobalamin acts as a coenzyme for methionine synthetase in re-methylation of homocysteine to methionine. This process also involves folate, which is transformed from 5-methyltetrahydrofolate (5-MTHF) to tetrahydrofolate (THF) by methionine synthetase (Hansen, Waagbø & Hemre, 2015). THF is important in cell division, thus folate deficiency also lead to megaloblastic anemia (Hansen, Waagbø & Hemre, 2015).

The increase in liver folate when diet levels increased above 1–2 mg kg −1 (requirement according to (NRC, 2011), is a first indication that Atlantic salmon diets should hold slightly higher folate levels than the present recommendations, when diets are based on plants (Hansen, Waagbø & Hemre, 2015). Folate act as a donor of one-carbon units in several biosynthetic processes. Deficiency results in loss of growth, which was also found in the present study with salmon parr in freshwater, when the NP addition was below 100NP and dietary folate levels below 2.25 mg kg−1. Similar to cobalamin deficiency, not enough folate results in hyperirritability, in addition to megaloblastic anemia (Ikeda et al., 1988). Hyperirritability was not observed in our study. No negative effects of folate levels as high as 10 mg kg−1 has ever been seen (Duncan et al., 1993). Folate has been extensively studied in other species in relation to embryo development and cancer (Said et al., 2000). NRC requirements lays between 0.6 and 1.0 mg kg−1 diet (NRC, 2011), in our study and based on the liver levels, folate in Atlantic salmon feeds are recommended to be 3.3 mg kg−1 or higher for both parr and post-smolt.

Tissue status of biotin was not affected by NP addition, showing that even the diet not added any NP, provided sufficient amounts to saturate tissue concentration. Biotin acts as a coenzyme for the carboxylases, e.g., acetyl-coenzyme-A carboxylase (ACCA) and pyruvate carboxylase (PyC), where it attaches to the amino group of specific lysine residues in these enzymes. Deficiency of biotin often results in damaging of organs, changed behavior and changed skin coloration (Hansen, Waagbø & Hemre, 2015). No such signs were seen in the present studies. However, very low biotin requirement is identified in some fish species, e.g., tilapia (0.05 mg kg−1; Shiau & Chin, 1999). The feed ingredients provided 0.26 mg biotin per kg diet. Expression of PCCA and PCCB (pyrivate carboxylase; two isoforms) in liver confirmed no effect of diet on the transcription level of this enzyme in the present study, except slightly lower expression of PCCB in the 400NP group. This provides a good reason to conclude that it is not necessary to add biotin to salmon diets based on the plant ingredients used in this study.

Niacin is part of the co-enzymes nicotine adenine dinucleotide (NAD), and NAD phosphate (NADP), transferring H+ and e− in the metabolism of carbohydrates, lipids and amino acids. However, as these are so widely participating in most metabolic reactions, no studies so far indicate that these enzymes can be used as biomarkers for requirement (Hansen, Waagbø & Hemre, 2015). Estimated niacin requirements vary from 7.4 to 150 mg kg−1 dry diet (Ng et al., 1997; Halver & Hardy, 2002). Deficiency will result in reduced growth, anorexia, abnormal swimming and haemorrhage, and reduced tissue concentrations (Hansen, Waagbø & Hemre, 2015). Whole body niacin increased over the whole range of supplementation. This can be translated to a niacin requirement above 400NP or 110 mg kg−1 dry diet. The significant and steady reduction in PPP-enzyme activity (6PGDH and G6PDH) might further support this. Our recommendation is therefore to add niacin at a level resulting around 65 mg kg−1 in the final diet, to secure growth in Atlantic salmon parr and post-smolt. This level exceeds current recommendations (1–10 mg kg−1, NRC, 2011).

Pantothenic acid, vitamin B5, transfers acetyl and acyl groups in energy metabolism. Mitochondria rich cells and cells subjected to high cell division, like gill tissue, are especially sensitive to deficiency of this vitamin (Olsvik et al., 2013). Increased proliferation has been registered in fish fed a diet with no panthotenic acid added, but was not investigated in the present study. Requirement for vitamin B5 for growth are reported from 6–23 mg kg−1 diet for different fish species (Hansen, Waagbø & Hemre, 2015). The sensitive gill tissue showed a steady increase in vitamin B5 concentration, from 2.3 to 5.9 mg kg−1 tissue, with a very high correlation to diet levels, in agreement with other studies on fish (Hansen, Waagbø & Hemre, 2015). To secure growth and sufficient gill concentrations, we recommend adding Atlantic salmon diets around 22 mg kg−1 diet, for both parr and smolt stages, which corresponds to the levels in the 150NP.

Conclusion

The change in diet ingredients where the majority of proteins and lipids come from plants will need adjusted micronutrient premix additions to secure optimal growth and metabolism. Due to faster growth, with a four-fold increase of weight in the parr stage, and doubling of weight in the post-smolt stage, our data are based on a short period of the production, but indications for both life stages are similar regarding body levels of the B-vitamins. Biotin and thiamine levels were sufficient in plant based diets, as no addition beyond the feed ingredients seemed to be necessary. The other B-vitamins are recommended to be added at or above NRC (2011) recommendations for salmonides to optimise growth, hinder change in liver lipid deposition, and saturate biomarkers specific for each vitamin. Therefore, based on current data updated recommendations for Atlantic salmon parr and post-smolt stages is a diet holding levels of niacin around 65 mg kg−1 diet (current NRC recommendation 1–10 mg kg−1, NRC, 2011), riboflavin 10–12mg kg−1 diet (NRC, 2011; 4–7 mg kg −1), cobalamin 0.17 mg kg−1 diet (NRC, 2011; 0.02 mg kg−1), folate 3.3 mg kg−1 diet (NRC, 2011; 1–2 mg kg−1), pyridoxine 10 mg kg −1 diet (NRC, 2011; 2–16 mg kg −1), and panthotenic acid 22 mg kg −1 diet (NRC, 2011; 20 mg kg−1). Please be aware of confounding effects due to the multi-approach design.

The two regression trials was a part of the EU-project ARRAINA; Advanced Research Initiatives for Nutrition and Aquaculture, (7th Framework Programme, FP7-288925, CP-IP Large-scale integrating project). Highly valuated is the technical assistance of Eva Mykkeltvedt (sampling and gene expression analyses), Jacob Wessels (PPP enzyme activity), the staff at the Nutrition department at NIFES, especially Nina Wollertsen and Emilie Lie (B-vitamin analyses), and Anita Birkenes (amino acid analyses). Technical staff at Matre Research Station for fish maintainance of the parr experiment, and Gildeskål for the post-smolt maintenance.

Additional Information and Declarations

Competing Interests

Author Contributions

Animal Ethics

Data Availability

Kristin Hamre is an Academic Editor for PeerJ, otherwise there are no competing interests. Joana Silva is an employee of BioMar, Norway, and has no competing interest or any conflict with this study and publication.

Gro-Ingunn Hemre, Kristin Hamre and Marit Espe conceived and designed the experiments, analyzed the data, wrote the paper, prepared figures and/or tables, reviewed drafts of the paper.

Erik-Jan Lock performed the experiments, analyzed the data, reviewed drafts of the paper.

Pål Asgeir Olsvik and Monica Sanden analyzed the data, wrote the paper, prepared figures and/or tables, reviewed drafts of the paper.

Bente Elisabeth Torstensen and Rune Waagbø conceived and designed the experiments, reviewed drafts of the paper.

Joana Silva conceived and designed the experiments, performed the experiments, contributed reagents/materials/analysis tools, reviewed drafts of the paper.

Ann-Cecilie Hansen performed the experiments, reviewed drafts of the paper.

Johan S. Johansen performed the experiments, contributed reagents/materials/analysis tools, reviewed drafts of the paper.

Nini H. Sissener performed the experiments, analyzed the data, wrote the paper, prepared figures and/or tables, reviewed drafts of the paper.

The following information was supplied relating to ethical approvals (i.e., approving body and any reference numbers):

Both feeding trials were conducted in accordance with Norwegian laws and regulations concerning experiments with live animals, which are overseen by the Norwegian Food Safety Authority. Permission for these specific experiments were given by the Directorate of Fisheries, and accepted for feeding trials at GIFAS, §13 (Akvakulturloven) and §28a (Lakseforskriften) (ref 13/11363), and acknowledged by the advisory board 27.11.12 (ref: ARRAINA regression trial permission).

The following information was supplied regarding data availability:

All data are included in the manuscript.

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
