# Peer review of "Atlantic salmon (Salmo salar) require increased dietary levels of B-vitamins when fed diets with high inclusion of plant based ingredients"

_PeerJ, doi:10.7717/peerj.2493_

## Round 0.1 · original submission · Minor Revisions

· Academic Editor

Minor Revisions

Your manuscript should be modified according to the suggestions given.

Reviewer 1 ·

Basic reporting

The overall presentation of this manuscript is very good in written. the paper provide sufficient rationale to address the hypothesis and objective of the study.

Experimental design

the experimental design clearly defined the research question and hypothesis. Data collection were well controlled and the methods were well described. However some minor revision are required as suggested in the text

Validity of the findings

The data of this manuscript was statistically sound and well controlled. The authors have developed a good interpretation on the parameters measured in this study and thus the conclusions are convincing.

Additional comments

This manuscript presents an important research addressing one of the issues challenging aquatic feed production when plant ingredients are used to replace fishmeal. The overall presentation is very good in quality. The data collection shows good support to the conclusions. There are some minor comments and suggestions need to be considered for a correction.

Annotated reviews are not available for download in order to protect the identity of reviewers who chose to remain anonymous.

·

Basic reporting

This is a well written manuscript representing a tremendous amount of work. I understand the concept of presenting the two trials in one paper, but the format of mixing each study into the same sections was a little confusing. The authors have a firm control of the literature and presented the case for the trials well.

Experimental design

This research adds important information to the literature for the development of alternative feeds for salmon. The explanation of why to evaluate vitamins, amino acids and cholesterol is understood, but would a better approach have been to evaluate these nutrient classes separately? The methods are well described. It was stated that dwarf males were eliminated from the samples, but was an analysis of the effect of diet on the number of dwarf males done before the elimination?

Validity of the findings

The regression curves for the vitamins are very clear. Why was a multiple range test used for the performance data instead of regression curves? With the approach of changing so many nutrients at the same time, a discussion of changing levels of one nutrient and its effect on fish growth or metabolism is highly confounded. The authors did appear to be careful to not to overstate cause and effect with language like “might have contributed to lower growth…..”, but I think acknowledging this confounding factor in the experimental design would be beneficial for the readers.

Additional comments

Line 90; NRC 2011?
Line 141; Font change
Line 151; Skretting ARC, is this a diet formula or a location?
Line 398; histidin
Figure 2; Tiamine = Thiamin?

---

## Round 0.2 · accepted · Accept

· Academic Editor

Accept

Thank you very much for improving your manuscript.